# Multimodal Structured Foundation Models for Noisy Documents: A 110M Encoder Where Structure Matches Scale on Clinical Key–Value Generation

**Yingyun Li**[1]  **Haiyang Qian**[*1]

## Abstract

Foundation models for structured data have so far assumed that the data arrives clean and tabular. In healthcare, however, a large share of structured records is locked inside scanned paper reports of heterogeneous layout (free-text notes, multi-column forms, and table-heavy lab and pathology reports) recovered through error-prone OCR. The standard pipeline turns these into rows of key–value pairs and treats the OCR step as upstream ground truth, even though its error rate varies systematically with capture quality and layout. We argue that this regime calls for a *multimodal* view of structured generation, in which OCR quality signals are treated as a first-class modality alongside text, and key–value structure is treated as a first-class pretraining target. We instantiate this view as a noise-aware encoder whose input layer fuses text with seven OCR-derived reliability signals, and whose pretraining replaces MLM/NSP with two structure-aware objectives over key–value pairs. On 3,582 OCR-derived clinical-report pages, an 110M-parameter model trained this way outperforms strong Chinese encoder baselines by up to $+4.6$ F1 points on end-to-end key–value pairing and beats a 0.6B medical decoder LLM by $+31.1$ F1 points — a result that runs against parameter-count-as-scaling-axis intuition. Treating noise as signal and structure as supervision is, in our setting, a stronger lever than scale, and we sketch the implications for tabular and document-grounded structured foundation models more broadly.

[1]AI Starfish, Hangzhou, China. Correspondence to: Haiyang Qian <haiyang.qian@aistarfish.com>.

*Proceedings of the $2^{nd}$ ICML Workshop on Foundation Models for Structured Data*, Seoul, South Korea. 2026. Copyright 2026 by the author(s).

## 1 Introduction

Foundation models for structured data are advancing rapidly on tabular and time-series benchmarks where the input arrives as clean, schema-aligned rows (Hollmann et al., 2023; 2025; Kim et al., 2024). In healthcare the structured layer does exist — clinical reports are organised as field headers (*keys*) followed by patient-specific content (*values*) — but it is hidden behind paper documents whose layouts and OCR errors are both heterogeneous. Recent document-aware VLMs (Wei et al., 2026) question even the raster-scan token order assumed by encoders, because real documents do not follow it. The structured-data-side analogue: how should models handle inputs whose structure is partially corrupted but accompanied by *machine-readable* signals about where the corruption is?

Two assumptions in the dominant paradigm clash with this reality. First, encoder-only and decoder-only models alike are pretrained on born-digital, clean text (Devlin et al., 2019; Lee et al., 2020; Alsentzer et al., 2019; Cui et al., 2020). They therefore underweight the OCR quality signals that every modern OCR engine emits for free — per-token confidence, character-break statistics, layout-alignment scores. Second, standard pretraining objectives (MLM, NSP, next-token prediction) are not aware of the key–value structure of clinical records. A model can recover the surface form of a sentence without learning that "creatinine" should be followed by a numeric value with a unit string.

In this paper, we take a different stance. We treat OCR noise as a *first-class input modality* (an OCR-derived reliability stream) alongside text, and we treat key–value structure as a *first-class pretraining target*. We use *multimodal* in a deliberately broad sense — text fused with a lightweight machine-readable reliability stream — rather than the text+pixels sense common in document VLMs: we fuse text with the per-token confidence, character-break, and layout-alignment statistics that OCR engines already produce, turning a discarded byproduct of the pipeline into a supervision signal. Pretraining is then driven by two structure-aware objectives, one over key span recovery and one over key–value matching, that mirror the downstream structured-generation task.

Each predicted $(k, v)$ pair is a (column, value) cell of the resulting *tabular EHR row* indexed by the page; the model output is therefore exactly the row format that a tabular foundation model would consume downstream.

We instantiate this view as an 110M-parameter Chinese encoder pretrained on a mix of OCR-derived clinical-report pages and clean medical text. On a 358-page held-out evaluation, the model outperforms strong Chinese encoder baselines by up to $+4.6$ F1 points on end-to-end key–value pairing and beats an instruction-tuned 0.6B Chinese decoder LLM by $+31.1$ F1 points — a result that runs counter to compute-optimal scaling intuition (Hoffmann et al., 2022).

We contribute (i) a multimodal-structured-FM framing of OCR-derived clinical key–value generation, (ii) a lightweight pretraining recipe combining a quality-aware input layer with two structure-aware KV objectives, and (iii) headline results on a real OCR-clinical-report corpus under which a $\sim$0.1B encoder competes with much larger generative models; full architecture, ablations, and interpretability analyses are deferred to an extended version. We use "foundation model" here in the FMSD sense of a pretrained encoder whose representations are intended to underlie multiple structured-data downstream tasks; the empirical evidence in this short paper is from a single clinical KV-extraction task, and we make no claim of broad cross-task transfer.

## 2 Related Work

**Foundation models for structured data.** Tabular foundation models such as TabPFN (Hollmann et al., 2023; 2025), TabICL (Qu et al., 2025), CARTE (Kim et al., 2024), and TabTransformer (Huang et al., 2020) aim to deliver in-context predictive performance across heterogeneous tabular tasks; analogous goals are pursued on time-series records, often with clinical EHR datasets such as MIMIC-IV (Johnson et al., 2023) as the canonical clean tabular resource. These models presuppose that the structured input is already available — rows aligned to a schema, columns carrying typed values. Real-world clinical pipelines violate this assumption at the *ingestion* boundary: the structured layer must be recovered from scanned clinical documents of mixed layout before any tabular FM sees it. This work is therefore complementary to tabular FMs: it studies the upstream multimodal generation problem whose output is exactly the kind of row a tabular FM consumes downstream.

**Domain-adaptive clinical encoders.** BioBERT (Lee et al., 2020) and ClinicalBERT (Alsentzer et al., 2019) adapt BERT (Devlin et al., 2019) to biomedical and clinical corpora through continued pretraining on PubMed and EHR notes. These models substantially improve biomedical entity recognition and clinical text understanding, but they are trained on *clean*, born-digital text and do not consume OCR-derived quality signals. Our work is complementary: we keep the encoder backbone familiar (e.g., MacBERT (Cui et al., 2020)) and instead inject the OCR-pipeline metadata that is discarded by these prior models.

**Document AI and structural priors.** Document foundation models (Xu et al., 2020; Huang et al., 2022) jointly model text and 2D layout, which is highly effective for forms with visual templates; OCR-free document VLMs such as Donut (Kim et al., 2022) take the opposite stance and skip the OCR step entirely. Recent work also questions the raster-scan token order assumed by such models (Wei et al., 2026), motivating semantically-driven visual-token reordering on the encoder side. Our choice keeps OCR but treats its quality signals as supervision, which is orthogonal to both directions. These approaches operate on the encoder; our work targets the *task side* — the structured-generation objective itself — by injecting per-token reliability signals already produced by the OCR engine and aligning the pretraining objective with the downstream key–value structure. Clinical reports in our setting do not follow a fixed visual template; the KV structure must be recovered from text and noisy layout signals. Recent clinical-IE benchmarks under realistic OCR conditions (Li et al., 2026) make this regime concrete and reproducible.

**Post-OCR correction.** A long line of work tries to clean noisy OCR text via spell checking or seq-to-seq correction (Nguyen et al., 2021; Rigaud et al., 2019). Such "repair" approaches treat noise as an artefact to be removed before downstream modelling, which discards useful information about which tokens the recogniser was unsure of. We instead propagate that uncertainty into the encoder's representations.

## 3 Approach

**Setting.** Each input is a single OCR-derived clinical-report page $d$ with token-level OCR metadata; the output is either a key set $\hat{K}$ (key discovery) or a set of pairs $\hat{S} = \{(\hat{k}, \hat{v})\}$ (end-to-end pairing). The latter is the row of a tabular EHR record indexed by the page, with each $\hat{k}$ drawn from an open vocabulary of clinical fields — structured generation whose target schema is the union of fields recovered across the corpus.

**Noise-aware input representation.** Standard transformer encoders combine token, position and segment embeddings. We introduce a fourth, OCR-derived embedding (a machine-readable reliability stream rather than a pixel-level modality) to expose per-token reliability — a free byproduct of the OCR engine — to attention from the very first layer. Concretely, the input representation is

$$E_{\text{input}}(x_i) = E_{\text{tok}}(x_i) + E_{\text{pos}}(p_i) + E_{\text{seg}}(s_i) + E_{\text{noise}}(v_i),$$

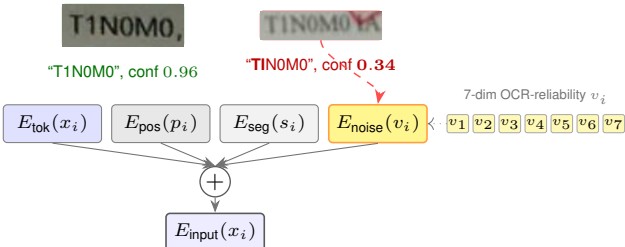

*Figure 1*. **Noise as a fourth embedding.** *Top:* two crops of the same TNM staging value "T1N0M0"; on the degraded scan (right) the OCR engine mis-recognises "1" as "I" with confidence 0.34, producing the clinically invalid "TIN0M0". *Bottom:* our input representation extends the standard BERT sum of token, position and segment embeddings with a fourth term $E_{\text{noise}}(v_i)$ (yellow), looked up from a 7-dim per-token OCR-reliability vector $v_i = (v_1, \ldots, v_7)$ (inset). Low-confidence tokens are routed to a dedicated noise bin so that attention can lean on TNM-staging context to recover the correct stage.

where $v_i$ is a low-dimensional vector of token-level OCR statistics (confidence summaries, character-break and layout-alignment indicators, etc.) produced by the OCR engine. The noise embedding $E_{\text{noise}}$ is intentionally lightweight: we discretise each component into a small number of bins and look up a per-dimension embedding, then sum. *Clean* inputs without OCR metadata (e.g., medical guidelines used as auxiliary pretraining text) are routed to a designated *anchor bin* in every dimension, so a single model can be trained on noisy and noise-free corpora simultaneously without distribution shift in the embedding space. The exact feature inventory and binning policy are deferred to the extended version.

**Structure-aware pretraining.** We replace the two original BERT objectives with structure-aware variants designed for clinical KV pages. (i) A *key-aware whole-word masking* objective extends WWM with a curated medical-entity and key-name dictionary as a *pretraining-time masking prior*; at inference the encoder still emits keys as free-text spans without vocabulary constraint, preserving the open-schema setting. (ii) A *key–value matching* objective replaces NSP with a binary classification over $(K, V)$ pairs, trained against *structurally* confusable negatives (e.g., a value paired with a key from a different clinical category, such as a lab measurement swapped onto a diagnosis field). Together they push the model to learn the directional, field-conditioned dependency that populates one row of the resulting tabular EHR.

**Compute footprint.** The model is a 12-layer, 768-hidden Chinese encoder ($\sim$110M params) pretrained on 537,721 sliding-window segments mixing OCR-derived clinical reports ($\approx$35%) with clean medical and general-domain text. The noise embedding adds negligible parameters and no measurable inference overhead.

*Table 1*. Headline results on real OCR-derived clinical reports. The proposed multimodal noise-aware encoder beats all encoder baselines and the two-shot 0.6B LLM, despite using only $\sim$110M parameters. $K_e/K_a$: key-discovery F1 (Task 1, exact / approximate). $K_eV_e$, $K_eV_a$, $K_aV_a$: end-to-end KV pairing F1 (Task 2); equivalently, tabular EHR row reconstruction F1.

| Model | Task 1 | | Task 2: KV Pairing | | |
|---|---|---|---|---|---|
| | $K_e$ | $K_a$ | $K_eV_e$ | $K_eV_a$ | $K_aV_a$ |
| Qwen3-0.6B (2-shot) | 0.058 | 0.060 | 0.381 | 0.420 | 0.422 |
| MBERT (0.18B) | 0.731 | 0.731 | 0.645 | 0.700 | 0.700 |
| RoBERTa-wwm (0.11B) | 0.735 | 0.735 | 0.636 | 0.692 | 0.693 |
| MacBERT (0.11B) | 0.734 | 0.735 | 0.646 | 0.705 | 0.706 |
| **Ours (0.11B)** | **0.768** | **0.768** | **0.692** | **0.717** | **0.718** |

## 4 Experiments

**Data.** We evaluate on 3,582 anonymised clinical-report page images collected with patient consent from an oncology corpus, digitised by a commercial OCR engine that emits both text and per-token quality signals. The evaluation set contains 358 pages with character-level KEY/VALUE annotations; the fine-tuning set contains 3,224 pages ($\approx$735 chars, $\approx$45,000 KEY annotations). Evaluation pages are page-level excluded from the 537,721 pretraining segments (no page-content leakage). We use the same corpus and split as the MedStruct-S benchmark (Li et al., 2026), a companion work on benchmark design; this paper contributes the noise-aware method on that split.

We evaluate two tasks: **Task 1 (Key Discovery)**: predict the key set $\hat{K}$ for a page; **Task 2 (End-to-End KV Pairing)**: predict the set of $(\hat{k}, \hat{v})$ pairs for a page (equivalently, a tabular EHR row).

**Baselines.** We compare against four strong Chinese encoders fine-tuned on the same data (BERT-Base-Chinese (Devlin et al., 2019), RoBERTa-wwm-ext (Liu et al., 2019), MacBERT (Cui et al., 2020), and an in-domain MBERT variant), and against an instruction-tuned 0.6B Chinese decoder LLM in a two-shot setting. All encoder baselines use an identical fine-tuning pipeline (BIO tagging head, shared optimiser, three runs).

**Metrics.** We report exact ($e$) and approximate ($a$, length-adaptive normalised edit distance) F1 over extracted spans; KV pairing uses three protocols $K_eV_e \subseteq K_eV_a \subseteq K_aV_a$ of decreasing strictness. For $K_eV_e$, each predicted key span is paired with its nearest following value span on the same page; this deterministic pairing is shared across all baselines.

**Headline result.** Table 1 and Fig. 2 show that the noise-aware encoder is the strongest model across all five metrics. Compared to MacBERT, end-to-end KV pairing improves by $+4.6$ F1 points ($K_eV_e$) and key discovery by $+3.4$ F1 points ($K_e$); compared to a 0.6B two-shot decoder LLM the encoder is roughly an order of magnitude smaller yet

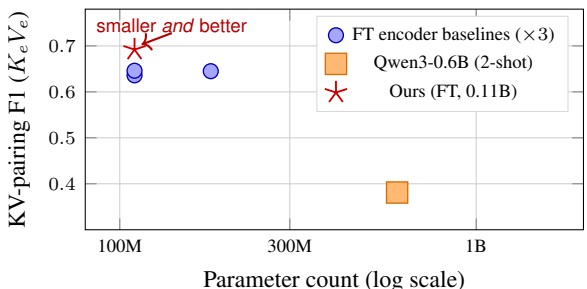

*Figure 2.* **Modelling OCR noise beats scaling.** End-to-end KV-pairing F1 ($K_eV_e$) versus parameter count on the same 358-page clinical-report test set. Our 110M noise-aware encoder (red star) outperforms three fine-tuned Chinese encoder baselines (blue circles: RoBERTa-wwm, MacBERT, MBERT) and a 5×-larger two-shot Qwen3-0.6B decoder LLM (orange square, lower right) by a wide margin. "FT" denotes supervised fine-tuning on the same training split; the LLM uses two-shot in-context learning only.

*Table 2.* Component ablation on the same test set as Tab. 1. Each row removes a single component from the full noise-aware encoder. $\Delta$ is the drop in end-to-end KV pairing $K_eV_e$ relative to Full.

| Variant | $K_e$ | $K_eV_e$ | $K_aV_a$ | $\Delta K_eV_e$ |
|---|---|---|---|---|
| **Full (ours)** | **0.768** | **0.692** | **0.718** | — |
| w/o Noise-Embedding | 0.752 | 0.680 | 0.704 | $-1.2$ |
| w/o KV-MLM | 0.752 | 0.683 | 0.709 | $-0.9$ |
| w/o KV-NSP | 0.755 | 0.689 | 0.710 | $-0.3$ |

$+31.1$ F1 points better on $K_eV_e$ ($0.381 \rightarrow 0.692$). **This headline gap conflates fine-tuning plus noise-aware structure with pure model size**; the companion MedStruct-S study (Li et al., 2026) on the same split reports that fine-tuned decoders up to 103B parameters still do not strictly dominate $\sim$0.1B encoders on $K_eV_e$, so the qualitative claim survives beyond our conservative two-shot 0.6B anchor.

**What drives the gain?** Tab. 2 reports component-level ablations: removing any single component degrades $K_eV_e$ by 0.3–1.2 F1 points, with the noise embedding the largest contributor. The gains are additive across all five metrics, and at clinical scale $\sim$1 F1 point on $K_eV_e$ corresponds to roughly one in a hundred cells of the predicted tabular EHR row. Full grid and faithfulness studies are in App. B.

**Case study: recovering "T1N0M0".** On the degraded scan of Fig. 1, all four encoder baselines emit unparseable stages from "**TI**N0M0"; ours down-weights the low-confidence sub-string, conditions on the valid "N0M0" suffix, and recovers (TNM stage, T1N0M0). The pattern generalises to other staging fields and recurrent unit errors (e.g., "μmol/L" → "umol/L").

**From KV-pairing F1 to tabular-row reconstruction.** Because each $(k, v)$ pair is one cell of the resulting tabular EHR row, $K_eV_e = 0.692$ doubles as a row-completeness metric: $\sim$69% of (column, value) cells in the predicted row exactly match ground truth. On a row of $n$ fields, $\approx 0.31n$ cells are

mis-filled, and a downstream tabular FM (Hollmann et al., 2023; 2025; Kim et al., 2024) consuming such rows inherits at least this much per-record label noise — bounding how much clinical tabular-FM error is recoverable by extraction quality alone, and making ingestion-quality variance a measurable lever for tabular-FM evaluation in healthcare.

## 5  Discussion and Conclusion

**Implication for multimodal structured foundation models.** Our results add a counterpoint to scaling-as-primary-axis at the document-grounded ingestion boundary: when the input pipeline already emits machine-readable reliability signals, a small encoder that consumes them and pretrains against the output structure can match much larger generative models on structured generation. The OCR-quality signal is then a third axis — alongside model and data scale — that is essentially free for any scan-originated deployment.

**Connection to tabular FMs.** Our pipeline outputs key–value pairs that are exactly the row format a tabular foundation model expects (Hollmann et al., 2023; Kim et al., 2024), so ingestion-quality variance propagates directly into any downstream tabular FM evaluation that uses such records. Treating that variance as a non-zero, structurally biased component of the pipeline — rather than as ground truth — is, we suggest, an underused calibration knob for tabular FMs in healthcare; at 110M parameters and on this task, modelling input noise and output structure jointly is a stronger axis than scale.

**Limitations and data statement.** Following reviewer suggestions: (i) the noise embedding is engine-specific — porting to Tesseract or engines lacking per-token confidence would require re-fitting the per-dimension bins, and a cross-engine study is left to an extended version; (ii) evaluation is on a single Chinese oncology corpus with no public document-AI benchmark comparison; (iii) single-run ablation deltas may fall within run-to-run variance; (iv) the LLM comparison, though bracketed above by fine-tuned 103B decoders on the same split, does not include a stronger *fine-tuned* in-context LLM of comparable size, which we plan to add. A unified-pipeline Qwen3-8B follow-up with Kendall-$\tau$ ordering and 3-tier clinical-risk analysis ($\sim$86% high-risk false-positive rate for the larger LLM) appears in an extended version. Reports were collected with patient consent; direct identifiers were removed before OCR and residuals filtered before training; the corpus is held on-premise with only aggregate statistics and anonymised examples reported.

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

# A    Noise-Embedding Feature Inventory

The noise embedding $E_{\text{noise}}(v_i)$ in the input representation of §3 consumes a 7-dimensional vector $v_i \in \mathbb{R}^7$ of token-level OCR statistics emitted by the recogniser. Each component is discretised into an independent learnable embedding via a per-feature binning function $\Phi_k$, and the seven embeddings are summed: $E_{\text{noise}}(v_i) = \sum_{k=1}^{7} \text{Lookup}\big(\Phi_k(v_{i,k}); W_k\big)$. Tab. 3 lists the seven features, their semantics, and the number of bins $N_k$ used in our experiments. *Clean* text without OCR metadata (e.g., medical guidelines used for auxiliary pretraining) is routed to a designated *anchor bin* ($\text{id} = 0$) in every dimension, so a single model can see noisy and noise-free corpora simultaneously without distribution shift.

| # | Feature | Description | $N_k$ |
|---|---------|-------------|-------|
| $v_1$ | $\text{conf}_{\text{avg}}$ | Average per-character confidence within the token | 64 |
| $v_2$ | $\text{conf}_{\text{min}}$ | Minimum per-character confidence (worst-case recognition) | 64 |
| $v_3$ | $\text{conf}_{\text{var}}$ | Logarithmic variance of confidence (recognition stability) | 32 |
| $v_4$ | $\text{conf}_{\text{gap}}$ | Confidence range $\text{avg} - \text{min}$ | 32 |
| $v_5$ | $\text{punct}_{\text{err}}$ | Punctuation error rate from rule-based artefact detection | 16 |
| $v_6$ | $\text{char}_{\text{break}}$ | Character-level fragmentation ratio | 32 |
| $v_7$ | $\text{align}_{\text{score}}$ | Vertical layout-alignment score (line and column coherence) | 64 |

*Table 3*. The seven OCR-quality features fused by the noise embedding. Bin counts $N_k$ were chosen by inspecting the empirical distribution of each feature on a held-out subset of the pretraining corpus and using quantile-based boundaries; values clustered near the high-end (e.g. confidence near 1.0) receive denser bins for higher resolution.

# B    Full Ablation Grid

Tab. 4 reports the full five-metric ablation grid referenced in §4. The headline pattern is the same as in Tab. 2: each component is non-redundant, and the noise embedding is the largest single contributor to end-to-end KV pairing. Two additional rows show the effect of replacing the bucketed noise embedding with a continuous projection (linear or MLP); the bucketing strategy outperforms both on $K_eV_e$, suggesting that the non-linear quantile-based discretisation captures aspects of the OCR quality distribution that a smooth projection misses.

| Variant | Task 1 | | Task 2: KV Pairing | | |
|---------|--------|--------|--------|--------|--------|
| | $K_e$ | $K_a$ | $K_eV_e$ | $K_eV_a$ | $K_aV_a$ |
| **Full (ours)** | **0.768** | **0.768** | **0.692** | **0.717** | **0.718** |
| w/o Noise-Embedding | 0.752 | 0.754 | 0.680 | 0.702 | 0.704 |
| w/o KV-MLM | 0.752 | 0.755 | 0.683 | 0.708 | 0.709 |
| w/o KV-NSP | 0.755 | 0.757 | 0.689 | 0.708 | 0.710 |
| Linear noise projection | 0.763 | 0.764 | 0.677 | 0.707 | 0.708 |
| MLP noise projection | 0.769 | 0.770 | 0.679 | 0.708 | 0.709 |

*Table 4*. Full ablation grid on the same evaluation set as Tab. 1. The bottom block compares discretised binning (used in Full) with continuous noise-feature projections.

# C    From a Page to a Tabular EHR Row

Fig. 3 illustrates the transformation that gives the title its name: a single OCR-derived clinical-report page is processed into a set of $(k, v)$ pairs, which together populate one row of a tabular EHR record indexed by the page. The schema is open-world — the union of clinical field names recovered across the corpus — so each row may have many NULL cells, and the row-completeness metric of §4 is the fraction of non-NULL cells whose value exactly matches the ground truth.

| Breast-cancer pathology report (excerpt) | Tabular EHR row produced |
|---|---|
| Specimen:  left modified radical mastectomy
Gross:  mass 2.5 × 2.0 × 1.8 cm, gray-white
Histologic type:  invasive ductal carcinoma
Histologic grade:  II (SBR)
TNM: pT2N1aMx
Lymph nodes:  3/15 metastatic
LVI: present   Margin:  negative
ER: + (90%)   PR: + (80%)
HER2:  2+   Ki-67:  25%  ⇒ | ```histologic_type    invasive ductal CA```
```histologic_grade   II```
```tumor_size         2.5 cm```
```tnm_stage          pT2N1aMx```
```ln_metastatic      3/15```
```lvi                positive```
```margin             negative```
```er                 + (90%)```
```pr                 + (80%)```
```her2               2+```
```ki67               25%```
```allergies          NULL``` |

*Figure 3.* One page in, one row out. A typical breast-cancer pathology report (the original is in Chinese; English shown here for readability) is processed into a set of $(k, v)$ pairs that populate one row of an open-world tabular EHR record. Fields not reported on this page (e.g., `allergies`) become `NULL` cells in the schema-aligned output; a downstream tabular foundation model that consumes these rows therefore sees the upstream extraction noise on every non-`NULL` cell.

