# OpenReview forum: "Multimodal Structured Foundation Models for Noisy Documents: A 110M Encoder Where Structure Matches Scale on Clinical Key--Value Generation"
_ICML.cc/2026/Workshop/FMSD — FMSD @ ICML 2026 Poster_

### Official Review · Reviewer_rJto · 2026-05-19
**Efficient multimodal encoder for clinical key-value extraction, but scope is too limited to be called a "Foundation Model"**

**Rating:** 7
**Confidence:** 4

**Review:**

**Summary**

This paper proposes a 110-million parameter multimodal encoder optimized for extracting key-value pairs from noisy, scanned clinical documents. The main innovation is in treating OCR errors and uncertainty metrics not as noise but as an additional input modality. The authors incorporate OCR engine confidence signals directly into the model's embedding layers and utilize pre-training objectives specifically tailored for key-value structures. The model demonstrates high efficiency, outperforming strong encoder baselines and a larger generative LLM (0.6B parameters) on a Chinese oncology dataset.

**Strengths**
* Propagating OCR confidence scores and error statistics directly into the input embeddings of the model is a clever approach that successfully bypasses the typical limitations found in traditional post-OCR correction methods.
*  Achieving impressive results with only $\sim 110\text{M}$ parameters while outperforming larger LLMs (such as Qwen3-$0.6\text{B}$) highlights that tailoring the architecture to structured data problems can be far more effective than blindly scaling model size.
* The construction of a pipeline that produces output directly in tabular formats compatible with downstream tabular models, facilitating integration with existing healthcare systems.

**Areas for Improvement**
* I strongly recommend that the authors consider using an alternative term instead of Foundation Model (e.g., Domain-Adaptive Encoder or Task-Specific Multimodal Encoder) both in the title and throughout the manuscript. While the term was introduced in the sense of the workshop idea, its usage here can lead to confusion. The evaluation presented is strictly empirical and restricted to a single clinical key–value extraction task, lacking evidence of generalization to other downstream tasks or diverse datasets, which is core to the definition of a foundation model. Unless further analyses are made, applying this term may overstate the scope and impact of the proposed approach.

*  Closely tied to the point above, a limitation is how the proposed framework is very dependent to one specific OCR engine. The noise embedding module was built to ingest the exact feature statistics generated by that particular engine. This raises questions about model's performance, if it can be severely degraded if switched to open-source alternatives like Tesseract, or if used on systems that lack these specific metrics? This heavy dependence diminishes the general utility of the approach.


**Detailed Comments**
*  The empirical validation is conducted entirely on a proprietary, on-premise dataset consisting of $3{,}582$ oncology reports. While privacy constraints in healthcare are understandable, the complete absence of evaluation on standard public benchmarks for visual/document information extraction makes it difficult to reproduce the work or fully verify the claims.

**Justification of Score**
The submission introduces an interesting and technically sound approach by merging OCR quality signals into a compact encoder's embedding layer, achieving excellent parameter efficiency. However, the impact of the paper is bottlenecked by an evaluation confined to a single proprietary task, and an architectural dependency on a specific commercial OCR engine. The work is promising and fits the workshop's theme well.

---

### Official Review · Reviewer_5VEM · 2026-05-20
**Review for "Multimodal Structured Foundation Models for Noisy Documents: A 110M Encoder Where Structure Matches Scale on Clinical Key--Value Generation"**

**Rating:** 7
**Confidence:** 3

**Review:**

# Summary

The paper extracts key–value pairs from OCR'd Chinese clinical pages with a 110M encoder, framing each page as one tabular EHR row. It adds OCR reliability as a fourth input embedding and replaces MLM/NSP with two structure-aware objectives: key-aware whole-word masking and key–value matching. On a 3,582-page on-prem oncology corpus, it beats Chinese encoder baselines by up to +4.6 F1 and a 0.6B two-shot decoder by +31.1 F1, arguing that noise-and-structure modeling beats scale here.

# Strengths

- The paper treats per-token OCR confidence as an input signal rather than discarding or repairing it, which is a clean and underexplored idea that adds negligible inference cost.

- It explicitly turns each page into one tabular EHR row, making the connection to downstream tabular models concrete rather than hand-waved.

- The evidence is internally coherent, with all baselines sharing one fine-tuning pipeline and split and an ablation that consistently isolates the noise embedding as the largest contributor.

- The method is lightweight and honestly scoped, with the authors explicit that the claim rests on a single KV task and no broad transfer is asserted.

# Areas for Improvement

- The "structure beats scale" claim is underpowered because the only large-model comparison is a 0.6B two-shot decoder, so the +31.1 gap largely reflects fine-tuning versus two-shot ICL rather than scale.

- Results come from a single dataset and a single OCR engine with no cross-engine test, even though the noise embedding is engine-specific.

- Leakage is unaddressed, since the eval set is a held-out subset of the same corpus used for pretraining but page-level exclusion is never described.

- The relationship to MedStruct-S is left unclear despite an identical metric set, which weakens reproducibility and positioning.

- No variance is reported, so several ablation deltas (0.3 to 0.9 F1) may fall within run-to-run noise.

# Detailed Comments

"Structurally confusable negatives" and the curated key dictionary are central but never operationalized, and the dictionary sits awkwardly with the stated open-vocabulary setting. The pairing/postprocessing step that KeVe actually scores is unspecified, and the TIN0M0 case study is suggestive but not systematic; a confidence-bin-stratified error analysis would support "noise as signal" far better. Efficiency claims would also benefit from concrete latency/memory numbers.

# Justification of Score

The idea is clean and the ingestion-boundary framing is a genuinely useful contribution, and the engineering is solid, so for a short paper I weight the novelty over experimental completeness. The headline scale comparison is underpowered and the dataset provenance and leakage handling need to be stated, but these are camera-ready-fixable and do not undermine the value of the result as something the workshop should see and discuss. I lean accept, and encourage the authors to clarify the MedStruct-S relationship and leakage handling, and either soften the scale claim or add a stronger fine-tuned decoder on the same split.

---

### Official Review · Reviewer_wLD2 · 2026-05-21
**Noise Aware Encoder for Key-Value Extraction**

**Rating:** 6
**Confidence:** 4

**Review:**

Summary: The paper adds OCR confidence signals as a fourth embedding to a BERT-style encoder and replaces MLM/NSP with two structure-aware pretraining objectives (key-aware masking, key–value matching). On 3,582 Chinese clinical report pages, the 110M model beats standard encoder baselines by up to +4.6 F1 on KV extraction and a 0.6B two-shot decoder LLM by +31.1 F1.

Strengths: The core idea is very rational. OCR models produce per-token confidence metadata that can be used as signal by  propagating it into the encoder. The TNM staging case study concretely shows how noise-awareness translates to clinically correct output where all baselines fail.

Weakness: The result comparison seems a bit misleading. The 0.6B LLM uses two-shot prompting while all encoders are fine-tuned on 3,224 pages. Single task, single language, single OCR engine might not generalize too well.